# Across-animal odor decoding by probabilistic manifold alignment

**Pedro Herrero-Vidal**
Center for Neural Science
Neuroscience Institute
New York University
pmh314@nyu.edu

**Dmitry Rinberg**
Neuroscience Institute
Center for Neural Science
NYU Langone Health
rinberg@nyu.edu

**Cristina Savin**
Center for Neural Science
Center for Data Science
New York University
cs5360@nyu.edu

## Abstract

Identifying the common structure of neural dynamics across subjects is key for extracting unifying principles of brain computation and for many brain machine interface applications. Here, we propose a novel probabilistic approach for aligning stimulus-evoked responses from multiple animals in a common low dimensional manifold and use hierarchical inference to identify which stimulus drives neural activity in any given trial. Our probabilistic decoder is robust to a range of features of the neural responses and significantly outperforms existing neural alignment procedures. When applied to recordings from the mouse olfactory bulb, our approach reveals low-dimensional population dynamics that are odor specific and have consistent structure across animals. Thus, our decoder can be used for increasing the robustness and scalability of neural-based chemical detection.

## 1 Introduction

Collective network dynamics are the foundation of neural computation, from early sensory encoding [1], to working memory [2], decision making [3], or motor control [4]. Neural population activity often has low-dimensional structure that is qualitatively preserved across sessions [3], behavioral states [5], and even animals [6–8]. Aligning neural datasets from multiple recordings and animal subjects into a common latent space could provide a powerful tool for extracting unifying principles of brain computations. Nonetheless, progress is hampered by the lack of robust statistical tools for extracting shared neural population dynamics across datasets.

The alignment of low-dimensional neural manifolds is equally important in brain-computer interface (BCI) applications, where the assumption of a common low dimensional structure in neural responses is used to compensate for instability in neural recordings and ensure robust performance over time [9, 10]. This is not only restricted to motor control; in the sensory domain, a new generation of chemical detectors exploit the unsurpassed sensitivity of rodent olfactory receptors, but bypass the limitations of training animals to report odor identity by directly decoding it from neural responses [11–13]. Unfortunately, the applicability of this idea is limited by the need to learn the mapping between neural activity and chemical identity on an animal-by-animal basis, which involves costly data collection [14]. Statistical tools that make use of data from previous animals to quickly calibrate the decoder in a new animal could dramatically increase the practical use of such technologies.

The basic structure of the rodent olfactory bulb (OB) is well-understood [15]: a vast number of different olfactory receptors project to distinct OB subpopulations (glomeruli) whose spatial organization is largely preserved across animals [16, 17]. Odor identity is encoded in low-dimensional transients of their population dynamics [1, 18, 13], though how the glomerular topography translates into stimulus-dependent network dynamics is less clear. Nonetheless, given the anatomical and functional similarity of the rodent olfactory bulb, it is likely that the OB response to individual odors is itself

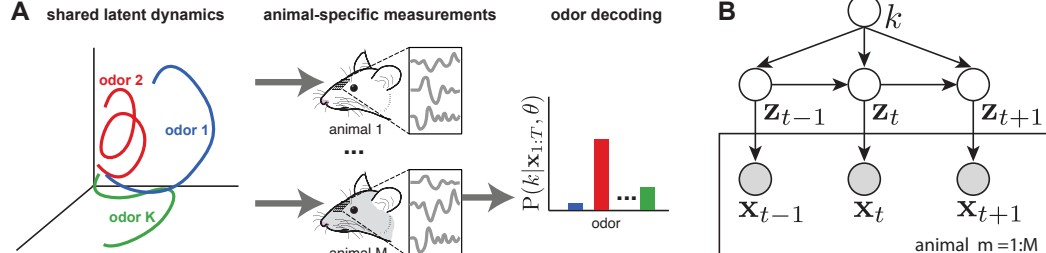

Figure 1: **Overview of the model. A.** Different odors elicit distinct trajectories in a shared neural latent dynamical space, with animal-specific measurements resulting in distinct observations. The goal is to decode the identity of a test odor, given parameters estimated from a dataset that combines measurements from all animals. **B.** amLDS graphical model with latent dynamics $\mathbf{z}_t$ specified by odor-specific $k$ parameters, and animal-specific observation model for measurements $\mathbf{x}_t$.

somewhat stereotyped across animals. With appropriate alignment, one could construct 'universal' odor decoders, applicable across animals.

Here we propose a novel probabilistic approach to aligning observations across neural recordings taken from different mice so as to extract commonalities in their circuit-level representations of different odors. Our estimation procedure, which uses our 'aligned mixture of latent dynamical systems' (amLDS) graphical model, learns independent mappings of different recordings into a common latent manifold.[1] In this space, neural trajectories are similar across animals but distinct across odors which allows for robust odor identification. Moreover, our probabilistic decoder provides uncertainty estimates which can be used for optimal decision making. We used extended numerical simulations to evaluate the properties of the decoder in response to a range of biologically relevant manipulations, such as variations in measurement noise, dimensionality and across-trial variability in the latent dynamics, (mis)alignment of electrodes across animals, etc. We then tested our procedure on recordings from the OB of mice in response to several monomolecular odors. When fit to this data, our model confirmed that encoding of odors relies on transient dynamics in a shared low-dimensional manifold and that these dynamics are largely preserved across animals. Taking advantage of this reproducible structure, we were able to obtain remarkable improvements in odor identification performance relative to alternative approaches.

## 2 Shared odor encoding dynamics across animals

We start by mathematically formalizing the idea of shared stimulus-specific neural dynamics with animal-specific measurements (Fig. 1A). The amLDS graphical model assumes $d$-dimensional latent variables $\mathbf{z}_t$, which evolve according to stochastic linear dynamics with odor-specific parameters (Fig. 1B):

$$\mathbf{z}_t = \mathbf{A}_k \mathbf{z}_{t-1} + \mathbf{b}_{k,t} + \mathbf{w}_t, \tag{1}$$

where $t = \{2 \ldots T\}$ indexes time within trial; $\mathbf{A}_k$ and $\mathbf{b}_k$ denote parameters specifying the dynamics in response to odor $k$, with independent noise, $\mathbf{w}_t \sim \mathcal{N}(\mathbf{0}, \mathbf{Q}_k)$. The prior for the initial conditions $z_1$ is normal, with zero mean and covariance $\mathbf{Q_0}$.

The shared odor-specific dynamics are mapped into measured neural responses via an animal-specific linear Gaussian observation model, shared across all odor conditions

$$\mathbf{x}_t^{(m)} = \mathbf{C}_m \mathbf{z}_t + \mathbf{v}_t, \tag{2}$$

with i.i.d. noise $\mathbf{v}_t \sim \mathcal{N}(\mathbf{0}, \mathbf{R}_m)$; $\mathbf{C}_m$ and $\mathbf{R}_m$ are observation model parameters specific to animal $m$. We will use $N_m$ to denote the size of observations for animal $m$. Note that the animal-specific observation model means that it is not strictly required for the neural measurements to be tightly matched. For instance, different animals may have different numbers of measurement channels (e.g. due to some electrodes becoming damaged). Moreover, the model accounts for variability in the SNR of different electrodes (across channels and devices). Assuming that the measurement process

---

[1]Code available: github.com/pedroherrerovidal/amLDS.

is consistent in terms of the properties of the electrodes and their alignment on the olfactory bulb surface, one can additionally define a prior that encourages $\mathbf{C}$s to be similar across animals. To keep the procedure general, we only consider the unconstrained version here.

**Parameter learning**

The dataset $\mathcal{D}$ combines measurements from $M$ animals, jointly covering all $K$ stimulus conditions. For each animal-stimulus pair, $(m, k)$, we have $I_{m,k}$ trials.[2] The goal of the parameter estimation procedure is to derive maximum likelihood estimates of all model parameters, jointly denoted by $\theta = \{\mathbf{A}_{1:K}, \mathbf{b}_{1:K}, \mathbf{Q}_{0:K}, \mathbf{C}_{1:M}, \mathbf{R}_{1:M}\}$. We use expectation maximization (EM) to optimize the parameters, by introducing the variational distribution $q(\mathbf{z})$ and using it to lower-bound the marginal likelihood [19]:[3]

$$\log \mathrm{P}(\mathcal{D}|\theta) = \sum_{i,m,k} \int_{\mathbf{z}} \log \mathrm{P}(\mathbf{x}|\theta) q(\mathbf{z}) \mathbf{dz}$$

$$= \sum_{i,m,k} \int_{\mathbf{z}} q(\mathbf{z}) \left( \log \mathrm{P}(\mathbf{z}, \mathbf{x}|\theta) - \log \mathrm{P}(\mathbf{z}|\mathbf{x}, \theta) \right) \mathbf{dz}$$

$$= \sum_{i,m,k} \left( \int_{\mathbf{z}} q(\mathbf{z}) \log \frac{\mathrm{P}(\mathbf{x}, \mathbf{z}|\theta)}{q(\mathbf{z})} \mathbf{dz} - \sum_{i,m,k} \int_{\mathbf{z}} q(\mathbf{z}) \log \frac{\mathrm{P}(\mathbf{z}|\mathbf{x}, \theta)}{q(\mathbf{z})} \mathbf{dz} \right)$$

$$= \mathcal{L}(q, \theta) + \sum_{i,m,k} \mathrm{KL}(q(\mathbf{z})||\mathrm{P}(\mathbf{z}|\mathbf{x}, \theta)).$$

Due to the Markovian structure of the dynamics, the joint probability of the latent and observations has a relatively simple joint Gaussian structure. The E-step optimizes $\mathcal{L}$ w.r.t. $q$, by setting $q(\mathbf{z}) = \mathrm{P}(\mathbf{z}|\mathbf{x}, \theta)$; this is computed by traditional Kalman smoothing using the appropriate trial-specific parameters (Suppl. S1).

The M-step optimizes $\mathcal{L}$ w.r.t. to $\theta$. The cost function decomposes into the sum of terms that only depend on the subject-specific observation parameters ($\mathbf{C}_m, \mathbf{R}_m$) and the condition-specific latent dynamic parameters ($\mathbf{A}_k, \mathbf{Q}_k, \mathbf{Q}_0, \mathbf{b}_{k,t}$), respectively, and can be optimized separately. Taking the derivative of the loss and setting it to zero results in parameter updates:

$$\mathbf{b}_{k,1}^{new} = \frac{1}{I_k} \sum_i \mathbb{E}[\mathbf{z}_1] \tag{3}$$

$$\mathbf{Q}_0^{new} = \frac{1}{I_k} \sum_i \left( \mathbb{E}[\mathbf{z}_1 \mathbf{z}_1^\top] - \mathbb{E}[\mathbf{z}_1]\mathbb{E}[\mathbf{z}_1]^\top \right), \tag{4}$$

where the indicator $i$ covers all trials of the same stimulus type, across all animals $I_k = \sum_m I_{m,k}$; expectations are taken under posterior for trial $i$ (explicit indexing omitted for brevity).

The same set of trials are used for updating the latent dynamics, and the input drive:

$$\mathbf{A}_k^{new} = \left( \sum_{i=1}^{I_k} \sum_{t=2}^{T} \left( \mathbb{E}[\mathbf{z}_t \mathbf{z}_{t-1}^\top] - \mathbf{b}_{k,t}\mathbb{E}[\mathbf{z}_{t-1}^\top] \right) \right) \left( \sum_{k=1}^{I_k} \sum_{t=2}^{T} \mathbb{E}[\mathbf{z}_{t-1} \mathbf{z}_{t-1}^\top] \right)^{-1} \tag{5}$$

$$\mathbf{b}_{k,t}^{new} = \frac{1}{I_k} \sum_{i=1}^{I_k} \left( \mathbb{E}[\mathbf{z}_t] - \mathbf{A}_k^{new}\mathbb{E}[\mathbf{z}_{t-1}] \right). \tag{6}$$

Finally, the updated noise covariance is computed as

$$\mathbf{Q}_k^{new} = \frac{1}{I_k(T-1)} \sum_{i=1}^{I_k} \sum_{t=2}^{T} \left( \mathbb{E}[\mathbf{z}_t \mathbf{z}_t^\top] - \mathbf{A}_k^{new}\mathbb{E}[\mathbf{z}_{t-1}\mathbf{z}_t^\top] - \mathbb{E}[\mathbf{z}_t \mathbf{z}_{t-1}^\top]\mathbf{A}_k^{new\top} \right.$$

$$+ \mathbf{A}_k^{new}\mathbb{E}[\mathbf{z}_{t-1}\mathbf{z}_{t-1}^\top]\mathbf{A}_k^{new\top} - \mathbb{E}[\mathbf{z}_t]\mathbf{b}_{k,t}^{new\top} - \mathbf{b}_{k,t}^{new}\mathbb{E}[\mathbf{z}_t]^\top + \mathbf{A}_k^{new}\mathbb{E}[\mathbf{z}_{t-1}]\mathbf{b}_{k,t}^{new\top}$$

$$\left. + \mathbf{b}_{k,t}^{new}\mathbb{E}[\mathbf{z}_{t-1}]^\top \mathbf{A}_k^{new\top} + \mathbf{b}_{k,t}^{new}\mathbf{b}_{k,t}^{new\top} \right). \tag{7}$$

---

[2]Note that it is not necessary that all stimuli are presented to all animals, nor that the number of stimulus presentations is identical across animals and conditions.

[3]To keep notation simple, we drop the explicit indexing by time, trial, stimulus type, and animal of $\mathbf{x}$ and $\mathbf{z}$.

The updates for the observation model are the standard Kalman ones, with the distinction that the index $i$ iterates across all trials and stimulus conditions for a particular animal (total number $I_m = \sum_k I_{m,k}$):

$$\mathbf{C}_m^{new} = \left( \sum_{i=1}^{I_m} \sum_{t=1}^{T} \mathbf{x}_t \mathbb{E}[\mathbf{z}_t] \right) \left( \sum_{i=1}^{I_m} \sum_{t=1}^{T} \mathbb{E}[\mathbf{z}_t \mathbf{z}_t^\top] \right)^{-1} \tag{8}$$

$$\mathbf{R}_m^{new} = \frac{1}{I_m T} \sum_{i,t} \left( \mathbf{x}_t \mathbf{x}_t^\top - \mathbf{C}_m^{new} \mathbb{E}[\mathbf{z}_t] \mathbf{x}_t^\top - \mathbf{x}_t \mathbb{E}[\mathbf{z}_t]^\top \mathbf{C}_m^{new\top} + \mathbf{C}_m^{new} \mathbb{E}[\mathbf{z}_t \mathbf{z}_t^\top] \mathbf{C}_m^{new\top} \right) \tag{9}$$

All the required posterior moments are obtained during the E step via Kalman smoothing as $\mathbb{E}[\mathbf{z}_t] = \mu_{t|T}$, $\mathbb{E}[\mathbf{z}_t \mathbf{z}_t^\top] = \mathbf{\Sigma}_{t|T} - \boldsymbol{\mu}_{t|T} \boldsymbol{\mu}_{t|T}^\top$ (see Suppl. S1 for further details).

As an important practical side note, the quality of the estimated parameter depends critically on a good initialization of $\theta$. We achieve this by fitting a simpler factor analysis model (FA), which ignores all temporal dependencies. This directly yields initial conditions for $\mathbf{C}_m$ and $\mathbf{R}_m$. The inputs $\mathbf{b}_{k,t}$ are initialized to the average of the corresponding inferred latent variable $\mathbf{z}_t$, using all trials from stimulus condition $k$, while parameters $\mathbf{A}_k$ and $\mathbf{Q}_{0:K}$ are initialized heuristically by linear regression using the FA-inferred posterior mean estimates of the latents.

We used Bayesian model comparison to determine the dimensionality of the latent space from data (evaluated on a separate validation set). As a more intuitive measure of goodness-of-fit, we also estimated the reconstruction error when predicting unobserved measurement dimensions (also known as leave-neuron-out error), commonly used for assessing the quality of models of neural population dynamics [20, 21]. More specifically, we use trials from the validation dataset, infer the posterior over the latent trajectory given data measurements in all electrodes except the $j$-th by Kalman smoothing $\mathrm{P}\left(\mathbf{z}_{1:T}|\mathbf{x}_{\neg j}, \theta\right)$, which is used to predict the values of the response of the $j$th neuron, $\hat{\mathbf{x}}_t^j = \mathbf{C}_m^j \mu_{t|T}^{(\neg j)}$, where $\mathbf{C}_m^j$ is the $j$th row of $\mathbf{C}_m$. We report reconstruction error as the squared distance between the reconstructed and measured activity, $\langle \|\mathbf{x}_t^j - \hat{\mathbf{x}}_t^j\|_2^2 \rangle_{j,i}$, where $\| \cdot \|_2$ is the L2 norm and $\langle \cdot \rangle_{j,i}$ is the mean over trials and choices for $j$.

### Odor decoding as hierarchical inference

At test time, one needs to infer the identity of the stimulus given the observations $\mathbf{x}_{1:T}$ in animal $M$, and the estimated parameters $\theta$, obtained by Bayes rule, $\mathrm{P}(k|\mathbf{x}_{1:T}, \theta) \propto \mathrm{P}(\mathbf{x}_{1:T}|k, \theta)\mathrm{P}(k)$. This requires estimating the marginal likelihood for each possible stimulus condition, which can be computed iteratively during the Kalman filtering procedure. More specifically, using the chain rule and then the Markov structure of the stimulus-specific LDS, we have

$$\mathrm{P}(\mathbf{x}_{1:T}|k, \theta) = \prod_{t=1}^{T} \mathrm{P}(\mathbf{x}_t|\mathbf{x}_{1:t-1}, k, \theta) = \prod_{t=1}^{T} \int \mathrm{P}(\mathbf{x}_t, \mathbf{z}_t|\mathbf{x}_{1:t-1}, k, \theta) \mathbf{dz_t}.$$

Taking the logarithm, rearranging the terms and taking advantage of the fact that the posterior marginals are normal with parameters obtained by Kalman filtering (see Suppl. Info. S1), $\mathrm{P}(\mathbf{z}_t|x_{1:t-1}, k, \theta) = \mathcal{N}(\mu_{t|t-1}, \Sigma_{t|t-1})$, yields:

$$\log \mathrm{P}(\mathbf{x}_{1:T}|k, \theta) = \sum_{t=1}^{T} \log \int \mathrm{P}(\mathbf{x}_t|\mathbf{z}_t, \mathbf{x}_{1:t-1}, \theta_k) \mathrm{P}(\mathbf{z}_t|x_{1:t-1}, k, \theta) \mathbf{dz_t}$$

$$= \sum_{t=1}^{T} \log \mathcal{N}\left(\mathbf{x}_t; \mathbf{C}_m \mu_{t|t-1}, \mathbf{C}_m \Sigma_{t|t-1} \mathbf{C}_m^\top + \mathbf{R}_m\right)$$

Lastly, here we assume a uniform prior over possible stimuli, to match the statistics of our OB dataset. Nonetheless, for real world applications the prior should reflect natural stimulus statistics.

## 3 Numerical experiments on simulated data

To validate the parameter estimation and decoding procedure, we constructed artificial datasets with the same statistical regularities assumed in the graphical model. More specifically, we defined a

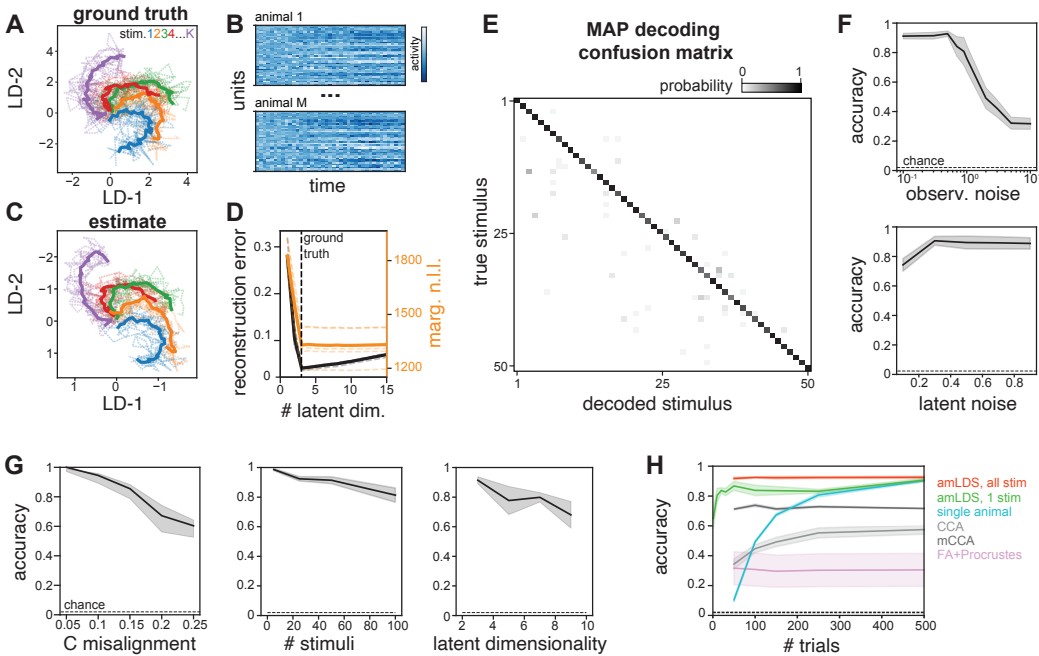

Figure 2: **Parameter estimation and decoding on simulated data**. **A.** Example latent dynamics for different odor stimuli; solid lines indicate the average latent trajectory and dashed lines are five random trials. **B.** Measurements of an example trial in two animals. **C.** Latents recovered with amLDS: dashed lines shows posterior mean of individual trials, solid lines show across-trials average. **D.** Reconstruction error as a function of assumed latent dimensionality, $d$ (ground truth $d = 3$). **E.** amLDS decoding performance measured using MAP estimates of the stimulus; $K = 50$. **F.** Decoding accuracy as a function of the magnitude of noise in the observations (top), and the latent dynamics (bottom). **G.** Decoding accuracy for parametric changes in model hyperparameters: alignment of axes across animals, $\mathbf{C}$ (left), the number of stimulus classes, $K$ (middle) and the dimensionality of the latent space, $d$ (right). **H.** Decoding accuracy as a function of the amount of data available for a target animal ($M = 5$) for amLDS when the target data covers all stimuli (red) or only one of the $K = 50$ stimuli (green), compared to using the same probabilistic model in only the target animal (blue), alignment by canonical correlation analysis (gray), mCCA (dark gray) and Procrustes alignment combined with factor analysis dimensionality reduction (pink). Solid lines show mean accuracy, shaded areas s.e.m. estimated across 5 independent experiments (random seeds).

shared low dimensional manifold ($d = 3$) and embedded $K = 50$ latent trajectories evolving over $T = 41$ time steps (Fig.2A). The stimulus-dependent inputs $\mathbf{b}_k$ were constructed using a common template with stimulus-specific amplitude (individual dimensions scaled by a value drawn from $\mathcal{N}(1; 0.0004)$) and rotation (evenly spaced over 170 degrees). The other latent dynamics parameters were set randomly: matrices $\mathbf{A}_k$ have diagonal entries drawn from $\mathcal{N}(0.4; 0.01)$ and off-diagonal drawn from $\mathcal{N}(0; 0.04)$ and the noise covariances $\mathbf{Q}_k$ and $\mathbf{Q}_0$ are diagonal with variances drawn from $\mathcal{N}(0.55; 0.0025)$. To model across-animal variability in the observation model we started from a prototype projection matrix $\mathbf{C}^*$, which we then morph into animal specific parameters $\mathbf{C}_m$ by corrupting the individual latent axes with independent additive noise $\mathcal{N}(0; \alpha^2)$, and rescaling their norm to a noisy value around $1.0$ (gaussian noise with variance $0.03$). Parameter $\alpha$ controls the degree of alignment of measurements across animals, with default value $0.1$. Finally, the observation noise covariance is set to be diagonal with variances set to the absolute value of independent draws from $\mathcal{N}(0; 0.25)$ (Fig.2B). This setup results in latent dynamics qualitatively similar to odor responses, and has the advantage of explicitly parametrizing the similarity across animals and stimuli.

As a first test of the estimator we simulated 5 animals, with $I_{m,k} = 50$ trials for each stimulus condition, which is on the same order of magnitude to the amount of data one may be able to collect in experiments. amLDS has low computational and memory demands; on a 2.9GHz CPU it takes 58 minutes for parameter learning (with 12500 trials; average over 5 runs) and 4 seconds to infer

stimulus class in a given trial (averaged over 2500 trials). In this setup, model comparison based on reconstruction error was able to correctly discover the dimensionality of the latent dynamics (Fig.2D) and reconstruct the underlying trajectories (Fig.2C), up to the expected degeneracies in the latent space scale and alignment [19]. Moreover, the decoding of stimulus identity of unseen test trials (20 per condition) revealed that hierarchical inference is able to correctly identify the true stimulus out of the 50 possible classes (Fig.2E), where we measure decoding accuracy as the fraction of test trials for which the MAP estimate of the stimulus class recovers the ground truth stimulus identity.

Qualitatively similar results can be obtained for a wide range of model choices. Performance is consistent across a wide range of noise levels in the observations (Fig.2F, top), which account for differences in recording quality. The same robustness can be measured for noise in the latent dynamics, due to potential changes in encoding owed to masking odors, variability in sniffing, etc ($\mathbf{Q} \sim \mathcal{N}(\mu, 0.0025)$, $\mu \in [0.1, 0.9]$; Fig.2F, bottom). The estimation is also relatively robust to different degrees of variability of the projections $\mathbf{C}_m$ across animals (Fig.2G, left), which may arise for example due to variability in the alignment of the electrodes on the OB surface. We also varied the dimensionality of the latent dynamics (Fig.2G, middle) and the number of stimulus classes (Fig.2G, right) and found that our decoding procedure shows robust performance for a wide range of settings. Given the relatively modest data requirements (50 trials per subject, and condition), these observations suggest that our approach is robust enough to be applicable to experimentally relevant data regimes.

To better quantify decoding performance as a function of the amount of data that needs to be collected in a new animal, we fixed the amount of data collected from 4 'source' animals ($K = 50$, $I_{m,k} = 50$) and varied the amount of new data collected in a fifth 'target' animal, for which we evaluated decoding accuracy. We found that in this setup amLDS yields high performance with a minimal amount of additional data (Fig.2F, red; one measurement per stimulus condition). Reaching the same decoding performance with data collected in the target animal alone would require many more trials, because the dynamics of all $K$ stimuli need to be learned de novo (Fig.2F, blue). In the case of amLDS however, (effectively) one only needs to align the new measurements to the manifolds estimated based on data from previous animals.[4] In fact, observing a few trials for a single stimulus is enough to achieve reasonable performance in the target animal (Fig.2F, green).

We compared the decoding performance of our method to three state-of-the-art methods: 1) across condition alignment by canonical correlation analysis (CCA) [22, 23] 2) multi-set CCA (mCCA) [24] and 3) Procrustes alignment with FA-based dimensionality reduction (FA+Procrustes) [9, 6]. In general, for all these approaches the latent dimensions are determined from data that combines all stimulus conditions, without explicit knowledge of the stimulus identity. Decoders are trained based on the resulting representation using stimulus labels.

CCA takes pairs of measurements from two animals and projects them into a shared low dimensional manifold so as to maximize the correlation between them. Since there is no meaningful way of pairing single trials, CCA requires averaging responses across trials within condition and pairing these average responses by stimulus type. Moreover, unlike our approach, both procedures treat measurements as independent over time, ignoring temporal structure in the neural responses. Since simple CCA can only be used for pairs of animals, we increased the number of trials in the single source animal to match the other methods. Additionally, we also used multi-set CCA, a generalization of CCA for more than two animals.

The alternative FA+Procrustes alignment is similar to CCA in that it requires across-trial averages and only considers similarity between pairs of measures; however, it can be applied multiple times to align animals $2 - M$ to the prototype axes given by animal 1 (the target animal data was used for FA and Procrustes parameter estimates in this case). More formally, we use orthogonal Procrustes [25, 9] where given a set of prototype $\mathbf{z}_{1:T}^{k*}$ and misaligned $\mathbf{O}\mathbf{z}_{1:T}^k$ trajectory pairs finds a rotation between the two, by minimizing the average misalignment after applying a rotation $O$, $\mathrm{argmin}_{\mathbf{O}} \sum_k \|\mathbf{z}_{1:T}^{k*} - \mathbf{O}\mathbf{z}_{1:T}^k\|_F^2$, under the constraint of orthogonality, $\mathbf{O}\mathbf{O}^\top = \mathbf{I}$; where $\| \cdot \|_F$ is the Frobenius norm. The stimulus-specific latent trajectories $\mathbf{z}_{1:T}^k$ are obtained via a dimensionality reduction step, performed independently in each animal by factor analysis. As there is no standard way for establishing the shared latent dimensionality based on data for (m)CCA and FA+Procrustes, we set it to the ground truth $d$ in all cases. Having established a linear map between observations and a common latent

---

[4]While the estimation procedure still uses EM based on the full data, the fractional contribution of the new measurements to the latent parameter updates is comparatively small, and becomes negligible as the number of sources increases.

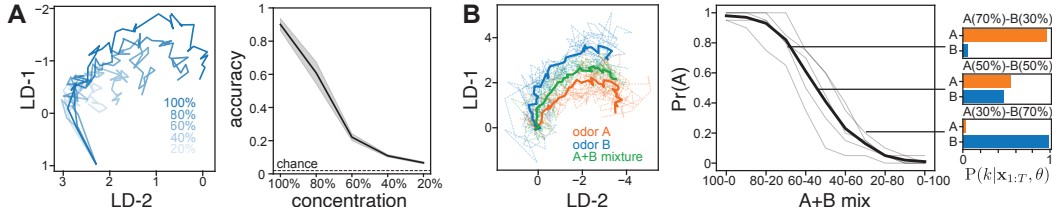

Figure 3: **Mixtures and concentration**. **A.** Estimation of odor identity across concentrations (right); example trajectories where decreasing concentration induces a compression of the stimulus trajectories (left). Shaded area shows s.e.m. across simulations. **B.** Example trajectories for two odors (A-orange and B-blue) and a 50-50% mixture that linearly interpolates between them (left). Frequency of the MAP estimate detecting odor A, as a function of the mixture ratio (middle) and three example posteriors (right); dashed lines show individual simulations, with average in solid line.

space by either procedure, individual trials across conditions were all projected in this space. We used a linear SVM with L2 regularization to decode stimulus identity from the latent trajectories [13, 26]. All controls were constructed using the library scikit-learn [27]. We found that although all three approaches performed above chance, none came close to matching the across animal decoding performance of amLDS.

**Robustness to fluctuations in concentration, odor mixtures**

Odor decoding in the wild needs to overcome further challenges that are not explicitly captured in the amLDS graphical model, such as fluctuations in concentration that rescale the amplitude and speed of odor specific responses [28]. Moreover, odorants are rarely present in isolation and have to be detected when presented in mixtures [29]. We used further numerical simulations to test how the estimator would respond in such scenarios.

We model across-trial variability in odor concentration as changes in the amplitude of the input drive $\mathbf{b}_{t,k}$ [30], with all other parameters the same as before. The parameter learning used data from the highest concentration and the decoder was applied to test trials with lower concentrations (Fig.3A). We found that the decoding accuracy degrades gracefully as the amplitude of responses decreases, remaining above chance for a wide range of concentrations.

It remains unclear how mixtures are encoded in OB responses [31], but a reasonable first approximation is to use a linear interpolation of sample trajectories for the individual odors present [32, 33]. For simplicity, we used $K = 2$ and further tied the parameters $\mathbf{A}$ and $\mathbf{Q}$ of the latent dynamics. We constructed two component mixtures which interpolate between stimuli labeled A and B (Fig.3B, left). Using a model trained on individual odors, we estimated decoding performance on mixture test trials while varying the ratio between the two components (Fig.3B, middle). The posterior over the two odors appropriately reflects the inherent ambiguity in the observations (Fig.3B, right). Overall, these results suggest that the decoding procedure is robust to naturalistic odor variations.

## 4   Across-animal decoding in the rodent olfactory bulb

We tested our model on neural recordings from a 64-site grid-electrode stereotaxically implanted over the dorsal part of the olfactory bulb in five mice [13]. We simultaneously recorded a pressure readout of the animal's sniff cycle. The animals were presented with five different monomolecular odorants (methyl valerate, ethyl tiglate, benzaldehyde, hexanal and background air), delivered in a temporally precise window (20-100 trials each). We extracted glomerular responses in a time window of 210ms from inhalation onset. The neural responses were preprocessed by removing the high frequency component (100Hz low-pass filter) and electrical noise (60Hz notch filter). Single electrode signals with peaks exceeding 2mV in a 5 second period were considered damaged and removed from the study. The electrode-specific signals were extracted by subtracting the instantaneous mean across all viable electrode sites.

We split the animals into 4 sources, and one target. All data from the source animals and $10\%$ of the trials from the target animal were used for parameter learning and hyperparameter estimation. The

remaining data from the target animal was used to assess odor decoding performance. When fitting the model to the neural recordings we found that the inferred latent trajectories were separable across stimuli and similar across animals (Fig. 4A). Some across-animal variability in the responses to the same odor would be expected in the model for a limited number of trials – due to the independent instantiation of the noise in the latent space, but it may also reflect unaccounted for sources of across-animal variability. The model comparison revealed a relatively small latent dimensionality in the data (Fig. 4B; we used $d = 7$ for all subsequent analyses). The model accurately captured variability in observed electrodes in the target animals, despite the very limited target data (2-6 trials per stimulus; Fig. 4C). The inferred map from the neural measurements into the latent space was similar across animals (Fig. 4D; see also Suppl. Fig. S1), likely reflecting the stereotypy of OB encoding and a precise alignment of the electrodes on the bulb surface.

We evaluated the decoding performance of amLDS on a target animal which revealed high accuracy MAP inference based decoding (Fig. 4E), with generally sharp single trial posteriors (Fig. 4F). We measured decoding accuracy for all possible choices of target animal and compared it to alternative estimators (Fig. 4G). The CCA, mCCA and FA+Procrustes estimation procedures were similar to those described for the simulated data, with the same latent dimensionality inferred for amLDS. We used across trial averages within each stimulus condition from the source animals and the 10% fraction of the target animal data for dimensionality reduction and across-animal alignment. We used the same data to train a linear SVM classifier based on the latent trajectories, which we then used to predict odor identity for the withheld target data. For CCA [22] we used single source animals and the training target data for the alignment and SVM training, averaging decoding performance on test data across all possible sources. Our probabilistic decoder significantly outperformed all three alternative alignment methods. For all possible choices of target, amLDS also systematically outperformed probabilistic decoding based on data from the target animal alone, confirming that pooling together data across animals improves data efficiency and overall decoding performance. While the single animal performance is close to that of amLDS given enough data, $82\% \pm 8\%$ vs. $76\% \pm 3\%$ when training with $50\%$ of the target animal's dataset (Fig. 4G), the difference between the two becomes dramatic in the low data regime used for amLDS ($10\%$ of target data used for training), with single animal performance going down to $60\% \pm 8\%$. This reinforces the idea that across-animal decoding is key to limiting the amount of data that needs to be collected for calibrating any new animal at deployment time.

## 5   Discussion

Animals are unparalleled in their ability to sense odors, which makes them an invaluable resource for a wide range of applications from security to medicine [34–36]. BCI approaches to odor detection attempt to bypass costly behavioral training by reading out odor information directly from the olfactory system, but they remain data limited [14]. Here we proposed a novel probabilistic procedure for mapping neural responses to odors into a shared manifold. This representation allows for robust odor decoding in a new animal with minimal amounts of further data collection. Calibrating the decoder to a new animal can be achieved with a small subset of stimuli, limiting the demand for rare/expensive chemicals. For instance, in our numerical experiments the decoder could extract odor identity in a new animal with as few as 2 trials for each stimulus, or 50 trials of a single stimulus. Moreover, we also found the estimation procedure to be robust to modeled fluctuations in odor concentration and the presence of background odors. These benefits were confirmed in data from mouse OB, where we found that decoding quality improved dramatically compared to previously-proposed alignment procedures, particularly when using a limited amount of data.

Our across-animal probabilistic decoding procedure brings neural chemical detection closer to practical applicability. Not only does it improve scalability and robustness, but it also significantly reduces data collection requirements. Furthermore, most odor detection applications have asymmetric costs and need to negotiate complex trade-offs between sensitivity and accuracy. For instance, when detecting *Clostridium difficile* [37], the medical costs of missing a positive diagnosis needs to be balanced against the financial costs of unnecessarily quarantining someone due to a false alarm. Our probabilistic decoder provides an explicit report of uncertainty associated with possible outcomes, which is critical for optimal decision making in these contexts.

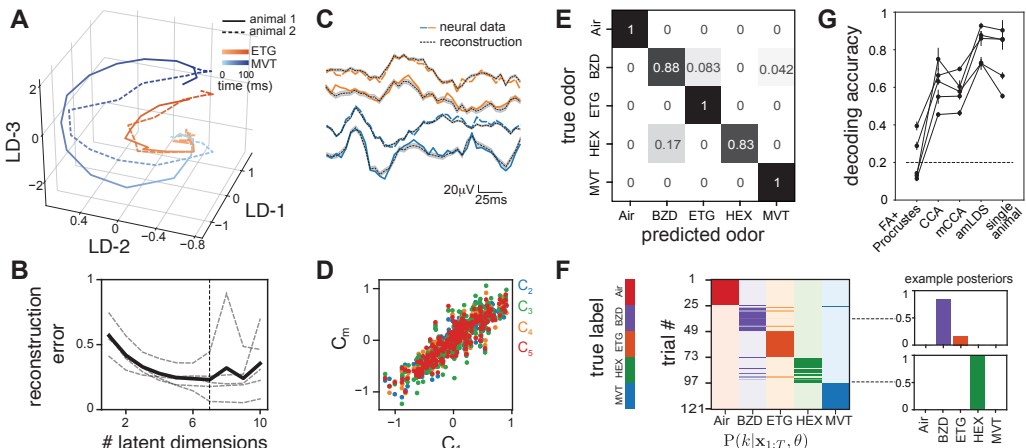

Figure 4: **Decoding performance on neural data from mouse olfactory bulb**. **A.** Inferred trajectories for two monomolecular odors (colors) in two animals (solid and dashed lines), averaged across 24 trials. **B.** Model selection based on reconstruction error performance; dashed lines show errors estimated for each target animal, solid line their average. **C.** Example odor-evoked responses and amLDS reconstructions for 2 stimuli and 2 animals. Shaded areas show 95% confidence intervals based on inferred posterior. **D.** Similarity of observation model across animals. **E.** Confusion matrix for amLDS decoding. **F.** Corresponding single trial posteriors; colors mark different inferred odors, trials are grouped by presented odor. **G.** Decoding performance for amLDS and alternative approaches based on CCA, mCCA, Procrustes alignment and a probabilistic decoder trained using data from the target animal alone (50% training, 50% test). Lines correspond to different choices for the target animal. Error bars show s.e.m. across 10 folds of the target data, for CCA s.e.m. across sources.

Alternative procedures for neural datasets alignment are usually not explicitly probabilistic.[5] Instead, they rely on a sequence of independent steps: first a (usually off-the-shelf) dimensionality reduction procedure. Dimensionality reduction is followed by across-condition data alignment within the latent manifold, using approaches such as CCA (and its multivariate generalizations [39, 40]), or Procrustes alignment [6, 41, 9]. Lastly, a common decoder is trained on the aligned across-condition data [22, 10, 9]. These procedures are popular as a replacement for costly and inconvenient BCI re-calibration in prosthetics [42], but their application to across-animal alignment remains rare [7, 6]. The multi-step procedure, although formally suboptimal, allows for some degree of flexibility in the individual component steps. For instance, the map into the latent manifold could be nonlinear, e.g. using Isomap [7, 43]. There also exist several nonlinear alignment procedures, such as kernel CCA or Distance Covariance Analysis, although such extensions have yet to translate into BCI practice [44]. In contrast, the hierarchical model proposed here is jointly estimated and linear in both the map and the latent dynamics. The linearity assumption here is not just a simplifying mathematical assumption, but it reflects domain-specific knowledge about the underlying neurophysiology of the system.

Neural activity has complex temporal dynamics and structured across-trial variability [45]. Previous approaches to aligning population activity across conditions via CCA or Procrustes alignment fail to capture either of these key statistical regularities. First, they treat the measurements across time as independent. Second, they intrinsically rely on paired measures, and since the correspondence of individual moments in time across trials in different animals is essentially arbitrary, they are restricted to across-trial average responses that may obscure important features of the underlying computation [46, 47]. It is also less clear how to determine the latent dimensionality of the data using these procedures. In contrast, amLDS extracts common low-dimensional dynamics using single trials, exploiting known features of the underlying computation. Moreover, the animal-specific observation model provides a principled framework for determining data dimensionality and for handling different sources of experimental variability (e.g. due to grid misalignment or missing data from defective electrodes). The probabilistic formulation also lends itself to a variety of generalizations. One could easily convert from Gaussian to Poisson observation noise for modeling spiking data instead of

---

[5]Some exceptions exist, such as probabilistic CCA [38], but they are not commonly used in practice.

the raw electric signals in our data [48, 49], or replace the linear dynamics with shared nonlinear equivalents [50, 51]. It should also be possible to extend the prior over trajectories to allow for some degree of heterogeneity across animals – for instance by allowing the parameters of the latent dynamics to slightly vary across animals, or to fall into a few distinct 'dynamics classes' (formally, an LDS mixture) [8]. Hence, amLDS can be thought of as yet another building block in the expanding statistical toolkit for understanding neural computation through the lens of dynamics in low-dimensional manifolds.

The representational stereotypy in the rodent olfactory bulb and the low-dimensional structure of its odor-induced dynamics are well documented [16–18, 13] make this system an ideal test case for out model. Nonetheless, we expect the same statistical tools to prove useful in other systems. The shared latent dynamics are best thought as reflecting a common computational solution, not a literal neuron-to-neuron, or subpopulation-to-subpopulation, match (in contrast to the hierarchical switching linear dynamical system approach of Ref. [8]). Hence, our model assumptions remain reasonable in circuits where representations are learned, rather than genetically encoded, as long as the dynamical system underlying the computation is low dimensional and the task strongly restricts the topology of the solution [52] (e.g. attractor dynamics for evidence integration [3]). We hope that our results inspire the search for common dynamic signatures in other brain areas and model systems.

**Author contributions.** CS and PHV conceived the original idea, developed the model and wrote the manuscript. PHV performed the numerical simulations and the data analysis. CS supervised the project. PHV collected the neural data in DR lab, previously published in Shor et al. 2022 [13].

**Declaration of competing interest.** DR is one of the founders of a company that commercializes related bio-electronic technology [13].

**Acknowledgements.** PHV is supported by training grant R90DA043849 (NIH). CS is supported by NIMH award 1R01MH125571-01, NSF award 1922658 and a Google research faculty award. DR was supported by DARPA BAA 15-35. We thank Caroline Haimerl, Edoardo Balzani and Erez Shor for their constructive feedback.

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
