# OpenReview forum: "Across-animal odor decoding by probabilistic manifold alignment"
_NeurIPS.cc/2021/Conference — NeurIPS 2021 Spotlight_

### Official Review · Reviewer_FTaj · 2021-07-16

**Rating:** 6
**Confidence:** 4

**Summary:**

The authors develop a method for aligning the neural activity of multiple animals. Specifically, they address the challenge of odor detection, where neural activity from multiple animals is recorded in response to many stimuli (odors), and then the odor needs to be decoded from a given animal’s neural activity. They use a hierarchical and probabilistic approach, in which there is a shared low-dimensional space of neural activity that is shared across animals, with different low->high dimensional mappings for each animal. The authors demonstrate the utility of the approach in simulations and on experimental neuroscience data.

**Limitations And Societal Impact:**

Limitations are addressed

Potential negative societal impacts are not mentioned in the text

**Main Review:**

Quality:

The quality of this paper is high. The submission is technically sound. The method was tested out rigorously in simulations and also shown to work well on experimental neuroscience data.


Originality:

This is a new method that builds upon linear dynamical systems / graphical modeling to create an effective model for the given problem.

The relevant literature was cited well.
One additional piece of relevant literature is a hierarchical switching linear dynamical system (which is similar to your approach but has an unknown k during training), as in
https://www.biorxiv.org/content/10.1101/621540v1.full


Significance:

I think the broad topic that the authors are trying to address - aligning neural activity across subjects has broad interest across the neuro/AI communities. However, the specific application of odor detection seems limited in terms of its broad appeal. I think this paper would be much stronger (and in my opinion, a clear accept rather than a borderline accept), if you were able to demonstrate its generality to other applications as well.


Clarity:

While much of the paper was written clearly, I found several aspects difficult to understand. In particular, I think it would be helpful to be more explicit in the beginning about what aspects of the model are learned during the training phase, and then what you are trying to predict during the testing/decoding phase (both for your model and the comparison models). Here are a couple specific points along those lines:

During training, is it assumed that the odor is always known? It seems like yes, but please clarify up front.

Lines 171-196: I initially found this section confusing since you describe decoding after describing the state-space alignment, so I was confused how you were actually doing decoding w/ CCA and FA+Procrustes. Having a sentence upfront about the overall approach would be helpful for clarity.

On your target animal, are you updating any of the parameters of the model besides learning the C and R for that new animal?

When decoding in the comparison methods, are you only using the target animal labels (odors) for the purpose of aligning the datasets, or are they also used when training the decoder?



Other:

I think multi-set CCA (which is an extension of CCA for >2 subjects) would be another natural comparison.

**Time Spent Reviewing:**

7

---

> ### Author Response · Authors · 2021-08-06
> **Reply reviewer FTaj**
>
> We greatly appreciate the reviewer’s feedback.
>
> We agree that expanding the analysis to more datasets would strengthen our claims and broaden the relevance of the work; at the moment, we don’t have access to the right kind of data from other brain regions (a reasonable number of animals doing the same task, with recordings from a circuit with low dimensional latent dynamics that are at least qualitatively similar across animals).
> The model implementation will be released with the manuscript, and hope for the community to start using and adapting our method to other datasets.
>
> We agree that Linderman et al. 2019  is a key reference and we will incorporate it in the updated version of the manuscript. However, it is important to note that their framework doesn’t aim to align recordings across animals. They were able to combine data from several *C. elegans* worms because of the genetic identity of the individual neurons (effectively the populations are pre-aligned), which allows for joint parameter learning.  In contrast, our generative model focused on aligning recordings across animals based on linear shared dynamics; to the best of our knowledge, ours is the only tools available for such probabilistic alignment. Nonetheless, it is true that the latent dynamics we are using are very simple and it may be important for future work to combine our approach with recent developments in latent space models that allow for non-linear latent dynamics, time wrapping or other observation models e.g. Duncker and Sahani, 2018, Linderman et al. 2019, Macke et al. 2015, Zaho and Park 2016.
>
> Clarifications about methodology:
>
> - The stimulus label is an observed variable at training time, and an inferred latent variable at test time (for decoding).  For the reported results, amLDS model parameters were trained using data from all of the source animals and some training data from the target animal. It would make sense in practice to use the training target data only for estimating C and R (the results for this version are qualitatively very similar). Training the animal-specific parameters also requires known stimulus labels, but not a complete stimulus set, and can be done using comparatively few trials.
> - When comparing to other methods, we used a fraction of the target animal’s data for both the alignment and decoder training.  We will spell these out more clearly in the final version of the manuscript.
>
> We will include the comparison to multi-set CCA in the final version of the manuscript.
>
> Our work doesn’t have any foreseeable negative societal impact, we will make sure to explicitly mention it in the final manuscript.

---

> > ### Comment · Reviewer_FTaj · 2021-08-26
> > **Thank you for your responses**
> >
> > Thank you for your responses and for clarifying your methodology in the final paper.
> >
> > I'm planning to stick with my initial score for the following reasons.
> >
> > I agree with reviewer tRLv that your model assumptions are somewhat limiting - in particular that there are identical dynamics for each animal. It could be nice if the dynamics parameters for each animal could vary slightly, and had a shared prior across animals (as in Linderman et al 2019). The assumption of identical dynamics makes me concerned that your specific model may not work very well on other applications/datasets.
> >
> > That being said, I am still relatively happy with the paper overall - the method works very well on the dataset within the paper, and it is the first probabilistic alignment paper that I'm aware of (so I don't think working on multiple datasets is an absolute necessity).

---

### Official Review · Reviewer_Qtbh · 2021-07-16

**Rating:** 7
**Confidence:** 3

**Summary:**

The report provides a method for decoding stimulus identity (odor) using data from multiple animals. The framework is straightforward and the performance appears respectable. I anticipate that this approach could be of use in neuroscience labs and could help with developing models and BCI. Given its general applicability, it would strengthen the impact if its use was demonstrated on spiking data and/or other types of stimuli or behaviors. This is a well written paper with helpful illustrations. I expect it would support interesting discussions.

**Ethical Concerns:**

I have no ethical concerns with this report.

**Limitations And Societal Impact:**

No concerns.

**Main Review:**

Neural activity is variable, even when sensory stimuli or motor performance are repeated. However, the underlying perceptions or intentions could be quite similar. When analyzing neural activity, it can be good to isolate the conserved components, despite this variability. A straighforward example is averaging over trials, but this only works when the dynamics are relatively time-locked. When the dynamics can play out over a range of speeds, the problem is less trivial. In olfaction, sync'ing to the sniff cycle is one way to register across trials, and in that case, trials are time warped to match a landmarks in a sniff cycle. For this paper, the author(s) developed an approach for solving this problem, and it is more general than timewarping to match the sniff cycle. It can be applied to a range of applications, sensory modalities, and behaviors. The framework allows decoding across animals. The approach is based on the idea that each animal has a set of neural activity manifolds that are each specific for a stimulus or behavior. Manifolds can be combined across animals for decoding. A graphical model, an aligned mixture of latent dynamical systems (amLDS), is computed from the dataset with this idea. They compare performance to CCA and FA+Procrustes. On the one hand, these are standard tools and so it is a reasonable comparison. On the other hand, they are very general tools that are used without taking advantage of the structure of the data, and so it is not so surprising that they provide less accurate results. The performance they present in Fig. 4 looks respectable.

Since they present olfaction data, I am curious to compare this approach to simple sync'ing to a sniff cycle (and they have that data, line 219). But I do not consider this a major weakness. I wish they had presented an application to spiking data. I will not mark that as bad weakness, but I do think it is something that others in the field would like to see. Relatedly, the impact could be greater if the technique were applied to a range of data types (e.g., visual stimuli, motor activity in a reaching task, parietal activity in a two-alternative forced-choice task). Again, it is not a weakness, but I would appreciate seeing it, and I bet I am not alone.

**Time Spent Reviewing:**

2

---

> ### Author Response · Authors · 2021-08-06
> **Reply reviewer Qtbh**
>
> We appreciate the reviewer’s positive reception of our work and its application to BCIs and the broader neuroscience community. We agree that the methods we compare against (CCA and FA+Procrustes) are ‘general purpose’ and do not exploit known structure of neural data. However, this is what the community actually uses at the moment, and, to our knowledge, ours is the only tool to probabilistically align neural recordings based on latent dynamics. We also agree that it is important to test the approach more broadly in other brain regions, although we don’t currently have access to the type of datasets that would make this possible. With appropriate preprocessing (Yu et al. 2009), our method can be used to spiking data, but to better account for spiking activity we could incorporate a Poisson observation model (which can be done in a straightforward manner, Macke et al. 2015). The model implementation will be made available with the final manuscript for the community to use and adapt to other problems and datasets.

---

> > ### Comment · Reviewer_Qtbh · 2021-08-25
> > **Nice extensions**
> >
> > I agree, those could be nice extensions, and would bring in a larger audience. Responses to the other reviews seem largely constructive. That's appreciated. I stand by my original score-- this is a good paper.

---

### Official Review · Reviewer_tRLv · 2021-07-16

**Rating:** 6
**Confidence:** 3

**Summary:**

Recent advances in recording techniques allow us to record from a large population of neurons - this high dimensional activity can typically be described by low dimensional latent factors. The authors leverage this low dimensional structure to capture the shared dynamical structure across animals in neural recordings from the olfactory bulb as animals are exposed to various odor stimuli. They fit a linear dynamical system (LDS) model with latent parameters and trajectories estimated for each odor that are shared across animals. These latent trajectories are subsequently mapped onto the neural recordings by employing animal specific observation parameters. The LDS model parameters are optimized using expectation maximization (EM). In order to decode the odor identities, the marginal likelihood for all stimulus conditions is computed. They validate the model on simulated data as well as neural recordings from the olfactory bulb.

**Limitations And Societal Impact:**

The authors have adequately addressed the limitations and potential negative societal impact of their work.

**Main Review:**

Strengths: Aligning low dimensional structure in neural population recordings across animals is an open problem in neuroscience. The paper provides a novel approach to recover these shared latent dynamics across animals. They demonstrate that their model can recover previously observed low dimensional transient dynamics for odor stimuli in data recorded from the olfactory bulb. They are also able to decode the odor identity across animals by training the model on source animals and a small subset of trials from a target animal. Moreover, the model outperforms CCA and Factor analysis followed by Procrustes alignment on across animal decoding.

Weaknesses: The authors make a strong simplifying assumption of identical dynamics across animals that has not been adequately justified in the paper.

Neural recordings during each session and across animals might be obtained from different locations in the same area or from a vastly different subset of neurons. It’s highly likely that the recorded neurons are part of different circuits and coding for different dynamical features.

Animals also have different sniffing cycles that could introduce an added level of complexity in the temporal dynamics that will not be accurately captured by this model. It is not necessary that the dynamics will be exactly temporally aligned.

Other factors such as varying levels of attention and arousal could further impact the dynamics of odor coding.

Due to the underlying assumption, this method is limited in scope and its applicability to other tasks and areas remains unclear. Maybe a weaker assumption with constraints on the dynamics parameters; for example, by employing a hierarchical model (Linderman et al., 2019), could make this model more flexible and as a result, more widely applicable.

Originality: To the best of my knowledge, this is the first work that aims to align neural dynamics across animals by having a common dynamics model.

Quality: The paper is well motivated and the writing and figures is easy to follow.

Clarity:
There are some minor parts that can be improved or might need further explanation.
Line 102: (typo) evaluated on on.
Line 77: It is not necessary to introduce a variational distribution since the authors are using regular EM.
Fig 4: I’m not sure I understand the source of variation between the latent trajectories for the same stimulus and different animals in panel A. Some further explanation here would be helpful.

Significance:
Aligning low dimensional dynamics across sessions and animals is elemental for identifying common encoding principles in the brain. However, as mentioned above, the strong assumptions in the model put stringent constraints on the dynamics and limit its applicability to other brain areas or contexts.



**Time Spent Reviewing:**

5

---

> ### Author Response · Authors · 2021-08-06
> **Reply reviewer tRLv**
>
> We thank the reviewer for noting the originality and relevance of our work and for suggesting ways to broaden its applicability.
>
> About the assumptions of the graphical model and how they may restrict the scope or applicability of this work:
>
> First, there is significant experimental evidence supporting the idea of representational stereotypy in the rodent olfactory bulb (Meister and Bonhoeffer 2006, Soucy et al. 2009) and that odors are decodable from low dimensional OB circuit dynamics (Bathellier et al. 2008). Moreover,  the data collection protocol was very similar across animals in our dataset, which means that the recordings come from the “same” or at least largely overlapping subpopulations, with the animal specific observation model accounting for any remaining neural misalignment. Lastly, the `behavior’ of the animals is quite consistent, w.r.t. arousal levels etc. Hence, the assumption of shared latent dynamics seems quite sensible for the particular application considered here.
>
> Second, we think of the latent dynamics as a shared computational solution, not a literal subpopulation or neurons match (in contrast to the hierarchical switching linear dynamical system approach of Linderman et al. 2019). As such, the same model assumptions should hold to some degree in circuits where representations are learned, rather than genetically encoded,  when the dynamical system underlying the computation is low dimensional and the task strongly restricts the topology of the solution (e.g. Mante, Sussillo et al. 2013 attractor dynamics for evidence integration) — this idea follows theoretical arguments about the universality of learned representations from Maheswaranathan et al. 2019 (and arguably if the dynamics are very different across animals then there is no meaningful notion of across-animal alignment to begin with).
>
> Third, it should be possible to extend the prior over trajectories to allow for some degree of heterogeneity across animals — for example allowing the parameters of the latent dynamics to vary somewhat across animals, or to fall into a few distinct ‘dynamics classes’ (formally, clustering the trajectories).
>
> Other issues:
>
> Our generative model provides the basic skeleton for across-animal alignment,  which can then be refined to include additional assumptions. For instance, if one finds the assumption of linearity of the latent dynamics too restrictive, it should be possible to incorporate nonlinear latent dynamics (using the work of Scott Linderman or Memming Park).
>
> Across-trial variability due to variations in sniff cycle length, attention/arousal etc are not part of the prior, but can be still captured at inference time, so such variability is taken into account by the decoder. It is also in principle possible to more formally account for variations in the speed of the dynamics (e.g. using time warping a la Duncker and Sahani, 2018).  Any of these additions require an extra level of hierarchical inference, and are left for future work.
>
> Panel A in figure 4 shows the estimated latent responses from different animals and stimuli in the shared latent manifold; trajectories cluster by stimuli (indicated by different colors) rather than by animal (indicated by different line styles); the plot shows condition-specific average trajectories, rather than individual trials.
>
> We will clarify all these issues in the final version of the manuscript.

---

> > ### Comment · Reviewer_tRLv · 2021-08-26
> > **Broader applicability**
> >
> > Thank you for the detailed response and clarifications. The assumption does seem to hold well in the case of olfactory bulb for this experimental setting as demonstrated by the impressive results. I have updated my score to reflect the same. I agree that identifying a shared topological solution is an important goal in understanding how neural circuits learn and implement tasks. However, that might not correspond to identical dynamics, especially in more complicated tasks. So I’m still not entirely convinced of its broader applicability.
> >
> > Overall, the idea of probabilistic alignment has a lot of potential and I’m excited to see its extensions on other datasets and modalities.

---

### Official Review · Reviewer_HVvC · 2021-07-16

**Rating:** 8
**Confidence:** 4

**Summary:**

This submission is a very well done application of the probabilistic inference to decoding of odour identity from neural recordings in mice. Interestingly, the latent code is assumed to have temporal dynamics shared across animals. The main benefits (high accuracy, data savings,  robustness to incomplete data) are shown in simulation and neural recordings. The method outperforms the existing (non-probabilistic) neural alignment procedures.


**Ethical Concerns:**

In terms of potential applications - is anyone considering using BMI for odour detection? Perhaps mentioning non-invasive recordings, or BMI improving life of human patients would be more convincing to a general audience?

**Limitations And Societal Impact:**

I assume the odours used in the experiment were very different. I'd be curious to learn about the temporal latent characteristics over more odours. What are the limits to discriminability? Is the detection of Clostridium difficile even feasible? I'm looking forward to updates into the "space of odours" following up this work.



**Main Review:**

To the best of my knowledge, this is a novel method for this important (BMI) application. It should have a large impact on the olfactory field, as it enables an effective combination of neural recordings from many animals.
The relevant work is adequately cited and discussed with two methods implemented for comparison.

The paper is clearly written and comprehensive.

Minor comments:
- L59: Shouldn't $x_1$ be $\mathbf{z}_1$?
- L119: Equation, first line is missing an integral over $\mathbf{z}_t$.
- Update Figure 4 B caption ("dashed lines...")

**Time Spent Reviewing:**

3

---

> ### Author Response · Authors · 2021-08-06
> **Replay reviewer HVvC**
>
> We appreciate the detailed feedback and will correct the typos.
>
> Regarding the limits of discriminability:  we could only investigate the large number of odors scenario in synthetic data, but we agree that the applicability of the approach warrants further experimental investigation; it will be important for future work to explore richer odor sets, with chemicals from the same and different chemical classes and complex mixtures of practical relevance. Mammals have been used in the detection of Clostridium difficile and other health conditions such as cancer or tuberculosis (Georgies et al. 2018, Seo et al. 2018), so presumably the required information is available in the olfactory bulb and therefore for the BMI. The wide range of applications, including but not limited to medical diagnosis, partly explains the recent interest in BMIs for odor detection (Dong et al. 2013, Saha et al. 2020, Shor et al. 2020).
>
> We expect that our method can be applied or extended to other BMI applications and the broader field of neuroscience/AI. The code will be released alongside the manuscript and we are excited for the community to use and adapt our method to other problems and datasets. We will highlight the broader range of applications in the updated version of the manuscript.

---

> > ### Comment · Reviewer_HVvC · 2021-08-25
> > **More about the potential impact of this research**
> >
> > Thank you for your response.
> >
> > Personally, I find the idea of using invasive BMI on animals controversial, particularly for large scale applications. However, I acknowledge that other venues have published such proposals (Dong et al. 2013, Saha et al. 2020, Shor et al. 2020).
> > What interested me in this research (somewhat justifying the use of animals) was the potential of extending our basic understanding of odour processing. What does the latent trajectory relate to? Does it correlate with perception? Does it define the limits of discriminability? Can it tell us about the "odour space" from the perspective of a given animal species? Would it differ more across species? Will your assumptions hold under naturalistic odour mixtures (cf Fig. 3)? Such questions would not be addressable without having a tool such as the one you propose, which is why I strongly support this submission.

---

### Decision · Program_Chairs · 2021-09-27

**Decision:**

Accept (Spotlight)

**Comment:**

This paper presents a novel probabilistic alignment of stimulus driven neural activity for decoding stimulus identity. Reviewers agree that this is an important contribution that can contribute to advancing neuroscientific understanding. At the same time, reviewers also agree on the assumptions that may limit its applicability outside the olfactory system. Please make sure to add the discussions through the rebuttal in the final version.